# The Role of Ultrasound in Diagnosing Endometrial Pathologies: Adherence to IETA Group Consensus and Preoperative Assessment of Myometrial Invasion in Endometrial Cancer

**DOI:** 10.3390/diagnostics15070891

**Published:** 2025-04-01

**Authors:** Mihaela Camelia Tîrnovanu, Elena Cojocaru, Vlad Gabriel Tîrnovanu, Bogdan Toma, Ștefan Dragoș Tîrnovanu, Ludmila Lozneanu, Razvan Socolov, Sorana Anton, Roxana Covali, Loredana Toma

**Affiliations:** 1Department of Mother and Child Medicine, “Grigore T. Popa” University of Medicine and Pharmacy, 700115 Iasi, Romania; mihaela.tirnovanu@umfiasi.ro (M.C.T.);; 2“Cuza Vodă” Women’s Clinical Hospital, 700038 Iasi, Romania; 3Department of Morphofunctional Sciences I—Pathology, “Grigore T. Popa” University of Medicine and Pharmacy, 700115 Iasi, Romania; 4Saint Josef Hospital, 65189 Wiesbaden, Germany; vlad.tirno@gmail.com; 5Department of Morphofunctional Sciences I—Histology, “Grigore T. Popa” University of Medicine and Pharmacy, 700115 Iasi, Romania; 6“Saint Spiridon” County Emergency Clinical Hospital, 700111 Iasi, Romania; 7Department of Surgery II—Orthopedics and Traumatology, “Grigore T. Popa” University of Medicine and Pharmacy, 700115 Iasi, Romania; 8“Elena Doamna” Women’s Clinical Hospital, 700038 Iasi, Romania; 9Department of Medical Bioscience, Faculty of Bioengineering, “Grigore T. Popa” University of Medicine and Pharmacy, 700115 Iasi, Romania

**Keywords:** ultrasonography, transvaginal ultrasound, color Doppler ultrasound, gray-scale ultrasound, endometrial pathology, endometrial cancer, myometrial invasion, IETA group criteria

## Abstract

**Background:** Ultrasonography is essential for diagnosing endometrial pathologies, such as hyperplasia, polyps, and endometrial cancer. The International Endometrial Tumor Analysis (IETA) group provides guidelines for using ultrasound to assess endometrial thickness, texture, and irregularities, aiding in the diagnosis of these conditions. The aim of this study was to evaluate the utility of various endometrial morphological features, as assessed by gray-scale ultrasound, and endometrial vascular features, as assessed by power Doppler ultrasound, in differentiating benign and malignant endometrial pathologies. A secondary objective was to compare the effectiveness of these ultrasound techniques in assessing myometrial invasion. **Methods:** A total of 162 women, both pre- and postmenopausal, with or without abnormal vaginal bleeding were enrolled in a prospective study. All participants underwent transvaginal gray-scale and color Doppler ultrasound examinations, conducted by examiners with over 15 years of experience in gynecological ultrasonography. Endometrial morphology and vascularity characteristics were evaluated based on the IETA group criteria, which include parameters such as endometrial uniformity, echogenicity, the three-layer pattern, regularity of the endometrial–myometrial border, Doppler color score, and vascular pattern (single dominant vessel with or without branching, multiple vessels with focal or multifocal origin, scattered vessels, color splashes, and circular flow). Sonographic findings were compared with histopathological results for comprehensive assessment. **Results:** The mean age of the study population was 56.46 ± 10.84 years, with a range from 36 to 88 years. Approximately 53.08% of the subjects were postmenopausal. The mean endometrial thickness, as measured by transvaginal ultrasonography, was 18.02 ± 10.94 mm with a range of 5 to 64 mm (*p* = 0.028), and it was found to be a significant predictor of endometrial malignancy. The AUC for the ROC analysis was 0.682 (95% CI: 0.452–0.912), with a cut-off threshold of 26 mm, yielding a sensitivity of 62.5% and a specificity of 89%. Vascularization was absent in 68.4% of patients with polyps. Among the cases with submucosal myomas, 80% exhibited a circular flow pattern. Malignant lesions were identified in 22.84% of the cases. Subjective ultrasound assessment of myometrial invasion, categorized as <50% or ≥50%, corresponded in all cases with the histopathological evaluation, demonstrating the effectiveness of ultrasound in evaluating myometrial invasion in endometrial cancer. **Conclusions:** In this study, cystic atrophic endometrium was identified as the most prevalent cause of postmenopausal bleeding. The most significant ultrasound parameters for predicting malignancy included heterogeneous endometrial echogenicity, increased endometrial thickness, and the presence of multiple vessels with multifocal origins or scattered vascular patterns. Additionally, color Doppler blood flow mapping was demonstrated to be an effective diagnostic tool for the differential diagnosis of benign intrauterine focal lesions.

## 1. Introduction

As the population continues to age and the obesity rate rises, there is an anticipated increase in estrogen-dependent endometrial pathology, including endometrial cancer. Consequently, clinicians need to be familiar with the current evidence and guidelines for the investigation and diagnosis of these conditions. It is therefore essential for gynecologists to know the physiological changes associated with menopause, as well as the typical ultrasound appearance of the endometrium throughout both the menstrual cycle and menopause [1].

Ultrasound became a key tool in diagnosing endometrial and intracavitary pathology in the late 1980s and early 1990s [2,3]. With the development of higher-resolution vaginal probes and color Doppler imaging, the diagnostic potential of ultrasound examination of the endometrium and the uterine cavity has increased substantially. This advancement has made ultrasound a powerful tool in estimating the risk of malignancy in symptomatic postmenopausal women, thereby aiding in optimizing patient management [4]. Several studies have been conducted to assess the clinical relevance of various patient characteristics or sonographic features [5,6].

Color and power Doppler ultrasound, as a non-invasive, simple, and quick procedure, can assist in distinguishing different endometrial lesions that can improve preoperative planning [7]. Furthermore, when B-mode ultrasound is integrated with color and power Doppler ultrasound as a non-invasive, expeditious modality, it often eliminates the need for additional invasive procedures to achieve an accurate diagnosis [8].

To standardize this ultrasound examination, the International Endometrial Tumor Analysis (IETA) group published a consensus paper outlining the methodology for examining the endometrium and uterine cavity, as well as the terms and definitions to be used when reporting ultrasound findings [9]. The adoption of standardized nomenclature between sonographers would promote more consistent communication, improve lesion description, and facilitate further studies on the application of this technology.

This investigation aimed to assess the diagnostic value of endometrial morphological features, as evaluated through gray-scale ultrasonography, and endometrial vascular features, as assessed via color Doppler ultrasonography, in differentiating between benign and malignant endometrial and intracavitary abnormalities. Furthermore, we explore the correlation between histopathological changes in the uterine cavity and sonographic observations in both pre- and postmenopausal females, with or without abnormal vaginal bleeding. A secondary objective was to identify the most reliable ultrasound modality for evaluating myometrial invasion.

## 2. Methodology

This prospective study focused on cases with endometrial pathology and was conducted between 1 January 2016 and 31 December 2021 at the “Cuza Vodă” Women’s Tertiary Clinical Hospital in Iași, Romania. A woman was defined as postmenopausal if she reported the absence of menstrual bleeding for at least one year after the age of 40, provided that the amenorrhea was not explained by medication or underlying disease. For this population, we included only those with abnormal uterine bleeding (AUB), without considering other premenopausal symptoms. Postmenopausal bleeding was defined as any vaginal bleeding occurring in a postmenopausal woman not receiving hormone replacement therapy. The inclusion criteria comprised premenopausal women presenting with abnormal uterine bleeding, any form of bleeding originating from the uterine cavity in postmenopausal women, or an increased endometrial thickness. The exclusion criteria for premenopausal women encompassed pregnancy, while for postmenopausal women, the exclusion criteria involved the use of hormone replacement therapy (HRT) or Tamoxifen treatment, and other potential causes of vaginal bleeding in postmenopausal women such as cervical cancer, cervical lesions, or vaginal lesions. This study included a total of 162 cases, of which 86 patients were postmenopausal and 76 patients were premenopausal. Among these, 146 cases were symptomatic, while 16 cases in the postmenopausal group were asymptomatic, characterized by a thickened endometrium observed via transvaginal ultrasound, but without fulfilling the criteria for endometrial cancer. All premenopausal cases were symptomatic. Of the cases diagnosed with cancer, 30 were postmenopausal and 7 were premenopausal (Figure 1).

The premenopausal women were evaluated for abnormal uterine bleeding unrelated to gestation. Additionally, the participant’s body weight status was evaluated. Following history taking, all patients underwent a physical examination, followed by standardized transvaginal ultrasonography without contrast enhancement, accompanied by color Doppler assessment.

Patients underwent transvaginal gray-scale and color Doppler ultrasound examination of the endometrium before undergoing an endometrial biopsy or hysterectomy. These assessments were conducted by examiners with over 15 years of experience in gynecological ultrasonography, utilizing a 5–9 MHz transvaginal transducer. All of the patients were examined in the lithotomy position with an empty bladder. The ultrasound procedure commenced with the acquisition of a proper midsagittal section of the uterus, followed by the measurement of the endometrium. In the presence of intracavitary fluid, both single layers were measured, and the sum was recorded. The uterus was scanned from right to left and from the fundus to the cervix. The endometrial morphology and vascularity were assessed, using IETA group criteria [9]: endometrial uniformity, echogenicity, three-layer pattern, regularity of the endometrial–myometrial junction, Doppler color score, and vascular pattern (single dominant vessel, with or without branching, multiple vessels with a focal or multifocal origin, scattered vessels, color splashes—large areas of color, and circular flow). The IETA color score, a subjective assessment of the endometrial blood flow, was recorded as follows: score 1 (no color), score 2 (minimal color), score 3 (moderate color), or score 4 (abundant color). Sonographic and histopathological results, obtained by uterine curettage or after hysterectomy, were evaluated. Endometrial curettage was performed in cases of abnormal uterine bleeding or when there was an increase in endometrial thickness observed in postmenopausal women. Hysterectomy, on the other hand, was performed exclusively for cases diagnosed with endometrial cancer. The pathologist was not blinded to clinical or ultrasound information. The histological endpoints included endometrial cancer, endometrial atrophy, endometrial hyperplasia, endometrial polyp, and intracavitary leiomyoma. The myometrial invasion was subjectively assessed as either <50% or ≥50% myometrial involvement (T1a or T1b). We did not utilize the full FIGO classification system, as our aim was to assess myometrial invasion specifically using transvaginal ultrasound as a diagnostic tool [10].

Statistical analysis was performed using the Statistical Package for Social Sciences for Windows (SPSS, Inc., Chicago, IL, USA, version 18) with a significance threshold set at 95%. For the statistical analysis, the following methods were used: Student’s *t*-test, Chi-square (χ^2^) test, Kruskal–Wallis test, and the area under curve (AUC). The Student’s *t*-test compared mean values between two groups with normal distributions, while the Chi-square test analyzed contingency tables for large sample sizes, focusing on frequency distributions. The Kruskal–Wallis test was employed to compare two or more groups for continuous or ordinal variables. Additionally, the receiver operating characteristic for endometrial pathology was used to balance sensitivity and specificity.

## 3. Results

We enrolled 162 women, of which 86 were postmenopausal and 76 were premenopausal. The sample size was determined based on the availability of eligible patients within the study period, as well as clinical feasibility. All the patients provided informed approval to participate in the study. The mean age of the participants was 56.46 ± 10.84 years (range: 36–88 years), with 53.08% of the subjects being postmenopausal. The clinical characteristics of the cohort are as follows: 90.25% of the women presented with symptoms, primarily abnormal uterine bleeding, while the remaining participants exhibited abnormal uterine findings without symptoms.

The average endometrial thickness, as measured by transvaginal ultrasonography, was found to be 18.02 ± 10.94 mm (range: 5–64 mm). Malignancy was identified in 22.84% of cases with uterine cavity findings. The mean endometrial thickness for women diagnosed with endometrial cancer was 24.49 ± 13.33 mm (95% confidence interval [CI]: 14.34 to 35.16), (Figure 2), whereas those with other pathologies had a mean thickness of 16.10 ± 9.37 (95% CI: 13.14 to 18.57), (Figure 3), with statistical significance (*p* = 0.028). When endometrial thickness was used as a standalone variable to build a model for estimating the risk for endometrial malignancy it demonstrated the highest AUC with a value of 0.682 (95% CI: 0.452–0.912). A cut-off threshold of 26 mm for endometrial thickness yielded a sensitivity of 62.5% and a specificity of 89% (Figure 4).

Among the patients with endometrial cancer, 81.8% were postmenopausal, with a statistically significant difference (*p* = 0.001). The mean age of participants with endometrial cancer was 61.97 ± 10.46 years, which was statistically significant (*p* = 0.001). All 37 participants diagnosed with endometrial malignancy presented with abnormal uterine bleeding, as reflected by the data (Table 1), which was a statistically significant finding (*p* = 0.003). The mean endometrial thickness in patients with endometrial cancer was 24.49 ± 13.33 mm, which was also statistically significant (*p* = 0.001). A hyperechoic mass was observed in 94.6% (*p* = 0.006), but all of the patients presented with an irregular endometrial–myometrial junction (*p* = 0.001). The mean Doppler color score in this cohort was 3 ± 1 (*p* = 0.001) and no patient demonstrated multiple vessels with focal origins or circular flow. Instead, 54.1% of the patients presented scattered vessels (*p* = 0.001). Furthermore, the study identified several factors that were significant in distinguishing malignant from benign intracavitary uterine pathologies, including menopausal status (*p* = 0.001), advanced age (*p* = 0.001), significant endometrial thickness (*p* = 0.001), an irregular endometrial–myometrial junction (*p* = 0.001), a high Doppler color score (*p* = 0.001), and the presence of scattered vessels (*p* = 0.001). A detailed comparison of these variables between patients with or without endometrial cancer is presented in Table 1.

In four cases of endometrial cancer, heterogenous echogenicity exhibited a sensitivity of 100% (95% CI: 0.6756 to 1) and specificity of 38.6% (95% CI: 0.2706 to 0.5157) (Figure 5). A hyperechoic appearance demonstrated a sensitivity of 50% (95% CI: 0.2152 to 0.7848) and specificity of 50.9% (95% CI: 0.3826 to 0.6338). The absence of a middle echo displayed a sensitivity of 100% (95% CI: 0.6756 to 1) and a specificity of 47.4% (95% CI: 0.3499 to 0.6008). An irregular endometrial–myometrial border proved to be an effective predictor of malignant lesions, with a sensitivity of 100% (95% CI: 0.6756 to 1) and specificity of 98.25% (95% CI: 0.9071 to 0.9969) (Figure 6 and Figure 7).

Regarding the color Doppler assessment, a vascular score of 1 usually excludes endometrial cancer, with a sensitivity of 87.5% and a specificity of 79% (Figure 8). This is due to the presence of a high number of newly formed vessels in malignant tumors (Figure 9).

Furthermore, color Doppler imaging proved valuable in more accurately determining myometrial invasion. The predominant feature of blood vessels in endometrial cancer was the presence of scattered vessels (Figure 10), which demonstrated a sensitivity of 50% (95% CI: 0.2152 to 0.7848) and a specificity of 94.74% (95% CI: 0.8563 to 0.9819). Additional vascular aspects indicative of endometrial cancer were the presence of vessels with various branches (Figure 11) and color splashes (Figure 12).

We accurately assessed the depth of myometrial invasion, greater or less than 50%, for all patients with endometrial malignancy (Figure 13 and Figure 14) using a subjective method. Furthermore, the depth of myometrial invasion was also verified preoperatively with magnetic resonance imaging (MRI). In each case of endometrial malignancy, our pathology department appreciated the invasion by conducting a frozen section during surgery, as well as through the final paraffin-based histopathological exam.

The mean age for patients with endometrial polyps was 50.70 ± 8.40 years (*p* = 0.001). The average endometrial thickness for the endometrial polyps was 13.68 ± 4.37 mm (*p* = 0.001), with a threshold value of 9.8 mm, yielding a sensitivity of 0.767, and specificity of 0.23, which lacked predictive value. There was no correlation between the internal structure of the endometrium and its echogenic homogeneity—heterogenous endometrium showed a sensitivity of 90.9% (95% CI: 0.7219 to 0.9747) and specificity of 46.51% (95% CI: 0.3251 to 0.6108) (Figure 15). The endometrial–myometrial junction remained regular in all cases of endometrial polyps. Usually, the vascular score for polyps was 1, with a single dominant vessel penetrating from the myometrium into the endometrium, either with (Figure 16) or without branching (Figure 17), although vascularization was not observed in 68.4% of participants with polyps.

A localized and consistent lesion extending into the uterine cavity is indicative of an endometrial polyp, with a sensitivity of 95.45% (95% CI: 0.782 to 0.9919) and specificity of 74.42% (95% CI: 0.5976 to 0.8507) (Figure 18), although it can also be observed in cases of submucosal fibroids and cystic atrophy of the endometrium (Figure 19).

In our study, the middle echo was present in 47.53% of cases, particularly among those with endometrial hyperplasia (20 cases), or endometrial polyps (57 cases). For the patients with endometrial hyperplasia, the threshold for endometrial thickness was 15.15 mm with uniform endometrial echogenicity showing a sensitivity of 66.7% and a specificity of 69.8%, alongside a hyperechogenic appearance. Among the patients with endometrial hyperplasia, 65% were premenopausal (*p* = 0.082), with a mean age of 50.10 ± 8.62 years (*p* = 0.005). Forty percent were obese (*p* = 0.321), and 85% were symptomatic (*p* = 0.438). The mean endometrial thickness was 15.15 mm ± 6.09 mm (*p* = 0.212). On ultrasound, 80% of the patients exhibited endometrial uniformity (*p* = 0.001), 50% presented with a hyperechogenic appearance (*p* = 0.001), and 90% had a regular endometrial–myometrial junction (*p* = 0.025). The color Doppler score was 1 ± 1 in most cases, with no branching vessels or circular flow observed. However, 35% of the cases showed multiple vessels with focal origin (*p* = 0.008).

Among patients with intracavitary myomas, 60% were premenopausal, with a mean age of 56.60 ± 12.30 years. The majority (80%) were of normal weight. Symptomatically, 60% of patients presented with symptoms (*p* = 0.068). The mean endometrial thickness was 16.80 mm ± 9.15 mm, and 60% of the cases exhibited endometrial uniformity (*p* = 0.172). A hyperechogenic endometrium was observed in 40% of cases (*p* = 0.061). Color Doppler imaging revealed a vascular score of 1 ± 1 in all cases (*p* = 0.928), with 80% of cases demonstrating a circular flow pattern (Figure 20). All patients had a regular endometrial–myometrial junction (*p* = 0.058). The cut-off value for endometrial thickness in identifying intracavitary myomas was 16.8 ± 9.15 mm, with a significant correlation between circular flow and the myomas (*p* = 0.001).

The mean age for cases with endometrial atrophy was 64.27 ± 8.32 years (*p* = 0.001). Among these individuals, 86.4% had a normal body weight (*p* = 0.001). Concerning clinical symptoms, 77.4% presented with postmenopausal vaginal bleeding. While endometrial thickness exhibited a high specificity for diagnosing endometrial atrophy, it demonstrated limited sensitivity. The average endometrial thickness was 16.78 ± 11.22 mm (*p* = 0.571). Gray-scale ultrasonography revealed heterogenous (*p* = 0.045) and hyperechogenic (*p* = 0.002) endometrial patterns, with a sensitivity of 100% (95% CI: 0.7847 to 1) and a specificity of 64.71% (95% CI: 0.5099 to 0.7637). In all cases, the middle echo was absent, and the endometrial–myometrial junction was regularly delineated (*p* = 0.001). For cystic atrophic endometrium, the vascular score cut-off was determined to be 2, yielding a sensitivity of 100% and a specificity of 86.3% (Figure 21).

## 4. Discussion

The clinical characteristics of our cohort showed that 90.25% of the participants presented with symptoms, primarily AUB, while the remaining participants had abnormal uterine findings without symptoms. This highlights the significant role of AUB in identifying intracavitary uterine pathologies in this population, emphasizing its prominence in clinical presentation. We compared our results with the findings from another study which reported that the most common symptom in perimenopausal women was menorrhagia (62.2%), along with additional symptoms such as pale skin and lower abdominal pain, and observed a similar trend. However, the study identified endometrial hyperplasia as the most prevalent pathology in both perimenopausal and postmenopausal groups (66.7% and 51.7%, respectively), whereas in our study, a larger proportion of women presented with AUB. This could suggest that a broader spectrum of pathologies was captured or that symptoms were more pronounced in our cohort [11].

Histological confirmation of intracavitary uterine pathology can be achieved through procedures such as dilatation and curettage (D&C), hysteroscopy, or outpatient sampling devices [12]. In our study, histological diagnoses were exclusively obtained via D&C for cases with abnormal uterine bleeding and through hysterectomy for patients diagnosed with endometrial cancer.

Numerous studies have investigated the role of endometrial thickness in the diagnosis of endometrial cancer [13,14,15,16,17]. In our cohort, the mean endometrial thickness was 18.02 ± 10.94 mm, which emphasizes the heterogenicity of endometrial pathologies enrolled in the study. However, the significant difference in mean endometrial thickness between women diagnosed with endometrial malignancy (24.49 ± 13.33 mm) vs. those with other pathologies (16.10 ± 9.37 mm) is noteworthy, although endometrial thickness alone is not a definitive diagnostic indicator. In our study, the endometrial thickness, as measured by transvaginal ultrasonography, plays a significant role in differentiating between endometrial cancer and other pathologies of the uterine cavity. This supports the growing body of the literature that suggests that increased endometrial thickness may be an important diagnostic marker for endometrial malignancy, especially in patients with abnormal uterine bleeding. Our study also highlights the importance of combining this ultrasound parameter with other diagnostic parameters such as the resistance index (RI), pulsatility index (PI), and peak systolic velocity (PSV) of the uterine arteries to further improve the differentiation between benign and malignant endometrial changes [18]. Therefore, additional ultrasound features are essential for differentiating between benign and malignant endometrial pathologies [19], a finding corroborated by our study.

Although our study was not designed to estimate the prevalence of intracavitary uterine pathologies in women with AUB, we did observe a higher prevalence of endometrial cancer cases (37 cases out of 162) compared to the existing literature. This increased prevalence may be influenced by the expertise of the physician performing the ultrasound, as the doctor in question specializes in oncological gynecological surgery. For endometrial cancer, the following ultrasound features were crucial for diagnosis: an irregular endometrial–myometrial junction, elevated color Doppler score, and the presence of scattered vessels aspect. These ultrasound findings have been shown to correlate with deeper myometrial invasion. Our findings support these characteristics as potential indicators of more advanced stages of disease, which could be significant for preoperative staging. Overall, irregular endometrial–myometrial borders and the detection of scattered vessels on Doppler ultrasound provide valuable diagnostic insights when evaluating individuals with suspected endometrial cancer. The use of these features in a systematic approach could potentially improve the accuracy of early detection and help in the decision-making process for further diagnostic or therapeutic interventions [17,20]. Dueholm M et al. (2019) emphasized the significance of a scoring system integrating both clinical and ultrasonographic assessments for evaluating the likelihood of endometrial malignancy in females with postmenopausal bleeding. This approach aims to expedite the diagnostic process and enhance the selection of appropriate second-line diagnostic or therapeutic strategies for women with elevated endometrial thickness [21]. Gheshev et al. demonstrated that 2DTVUS is a reliable and simple method for assessing myometrial invasion and cervical stromal involvement in endometrial cancer, with acceptable sensitivity and specificity [22]. Our findings align with these results, as we observed similar reliability in subjective ultrasound for determining the depth of myometrial invasion. Further supporting the utility of subjective ultrasound, Fruhauf et al. found that it performed better than objective methods (such as Gordon’s ratio and Karlsson’s ratio) in most cases, with a statistically significant improvement in sensitivity [23]. In their study, subjective assessment had a sensitivity of 79.3%, a specificity of 73.2%, and an overall accuracy of 75.7%. While subjective ultrasound in our study did not include these ratios, we also found that it effectively evaluated myometrial invasion, reinforcing its reliability as an accurate method for this purpose. In addition, Cerovac et al., examined the diagnostic accuracy of subjective TVUS in evaluating myometrial invasion, reporting excellent results. The kappa concordance coefficient between subjective TVUS and histological findings was statistically significant (*p* < 0.001) with a value of 0.72, suggesting good agreement between the two methods. Moreover, their study reported high diagnostic accuracy for detecting myometrial invasion greater than 50% (accuracy = 0.87, sensitivity = 0.77, specificity = 0.94) [24]. These findings align with our results, indicating that subjective ultrasound is a highly valuable tool for assessing myometrial invasion, with comparable sensitivity and specificity to histopathological evaluation.

Regarding the endometrial polyps, the endometrial–myometrial junction was consistently regular, with a vascular score of 1, and a single dominant vessel penetrating from the myometrium into the endometrium. Kucur KS et al. demonstrated that a single dominant or branching single dominant vessel pattern had high specificity for diagnosing endometrial polyps (98.28%), though its sensitivity was lower (66.67%) [25]. Interestingly, in our study, 68.4% of patients exhibited no vascularization, suggesting that a significant portion of polyps may lack the expected vascular patterns.

The ultrasound features with statistical significance for endometrial hyperplasia in our study were endometrial uniformity (*p* = 0.001) and a hyperechogenic aspect (*p* = 0.001). A circular flow pattern (*p* = 0.001) was the sole ultrasound feature indicative of intracavitary myomas. For cystic atrophic endometrium, the middle echo was absent, the endometrium appeared hyperechogenic (*p* = 0.002), and a regular junction was present (*p* = 0.001). The cut-off for the vascular score was 2, with a sensitivity of 100% and specificity of 86.3%. Ultrasonography helped distinguish endometrial hyperplasia from endometrial malignancy, particularly through the measurement of endometrial thickness using gray-scale imaging. Additionally, the application of color Doppler imaging enabled the detection of the blood flow within the lesion, further aiding in the differentiation between benign and malignant endometrial pathologies [26].

A prospective multicenter study employing the IETA terminology—which delineates the sonographic features of intracavitary pathology in pre- and postmenopausal females without abnormal uterine bleeding—provided valuable insights into the sonographic presentation of endometrial cancer. This study found that in asymptomatic women, endometrial cancers typically exhibited lower vascularity, with a predominant single vessel often observed either with or without branching. Additionally, these cancers are more commonly associated with a thinner endometrial lining and a more frequently regular endometrial–myometrial junction in comparison to women presenting with abnormal uterine bleeding [13]. These findings suggest that asymptomatic endometrial malignancies may present with distinctive ultrasound findings, including reduced vascularization and a more uniform junction, which could aid in differentiating malignant from benign pathologies in women without clinical symptoms [15,16,27]. However, our study was unable to replicate these observations because all patients diagnosed with endometrial cancer in our cohort presented with abnormal uterine bleeding. This homogeneity in clinical presentation precluded a comparison of sonographic features between asymptomatic and symptomatic cases. As a result, the potential role of specific ultrasound characteristics in the early detection of endometrial cancer in the absence of clinical symptoms could not be assessed. This underscores the necessity for further studies that include both symptomatic and asymptomatic cohorts to further elucidate the relationship between clinical presentation and sonographic findings, thereby enhancing the diagnostic accuracy of ultrasound in the evaluation of endometrial cancer. In our study, the key ultrasound parameters for predicting endometrial malignancy included heterogeneous endometrial echogenicity, endometrial thickness exceeding 24 mm, an irregular endometrial–myometrial junction, elevated vascular score, and the presence of scattered vessels.

We acknowledge that there are limitations in our study, particularly regarding sample selection and potential biases. Specifically, as noted, the timing of the sonographic examination in premenopausal women was not standardized to the early proliferative phase (between days 4–6) [9], which may have introduced variability in the results. Our participants were examined at the time of presentation, which could have influenced the consistency of the ultrasound findings. We also recognize the controversial nature of surgical management for asymptomatic endometrial polyps, especially in cases of AUB. As such, we only included cases of endometrial polyps that were confirmed by histological results. Additionally, while the number of malignant cases in this study was 37, we agree that a larger sample size would provide more robust data, particularly in evaluating the accuracy of ultrasound in detecting myometrial invasion. We plan to address this limitation by assessing a larger cohort in future studies.

## 5. Conclusions

In our study, the most common cause of postmenopausal bleeding was atrophic endometrium and cystic atrophy. Ultrasound features that most effectively predicted malignancy included heterogeneous endometrial echogenicity, endometrial thickness greater than 24 mm, irregular endometrial–myometrial junction, high vascular score, and scattered vessels. Color Doppler blood flow mapping proved to be a valuable diagnostic tool in distinguishing between benign intrauterine focal lesions. The IETA group nomenclature is clinically valuable and reasonable; the use of standardization of nomenclature between sonographers allows more consistent communication and better-described lesions.

## Figures and Tables

**Figure 1 diagnostics-15-00891-f001:**
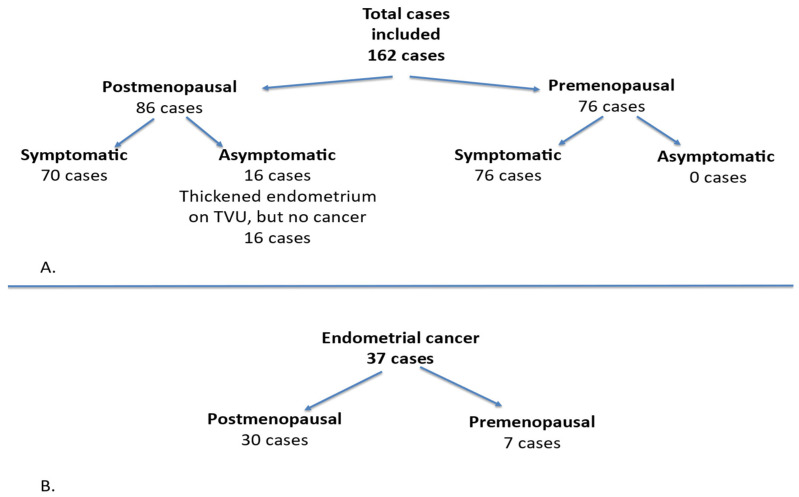
Patient enrollment and categorization. (**A**) Distribution of total cases; (**B**) distribution of patients with endometrial cancer.

**Figure 2 diagnostics-15-00891-f002:**
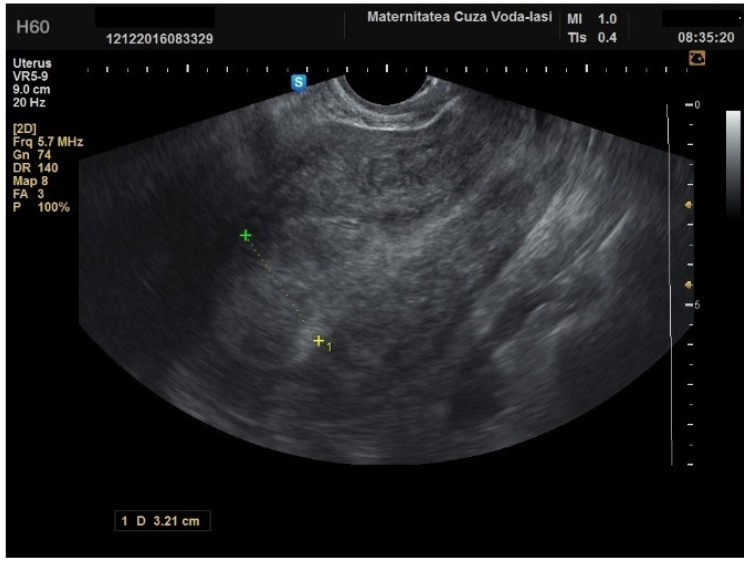
Ultrasonographic imaging of endometrial cancer.

**Figure 3 diagnostics-15-00891-f003:**
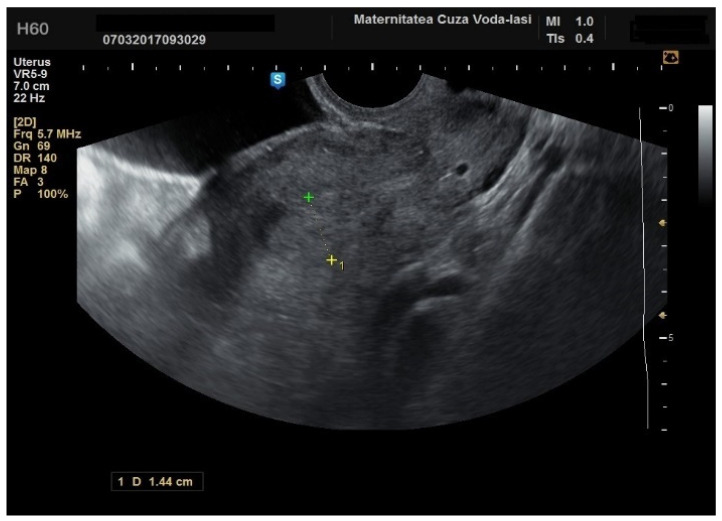
Ultrasonographic imaging of endometrial polyps.

**Figure 4 diagnostics-15-00891-f004:**
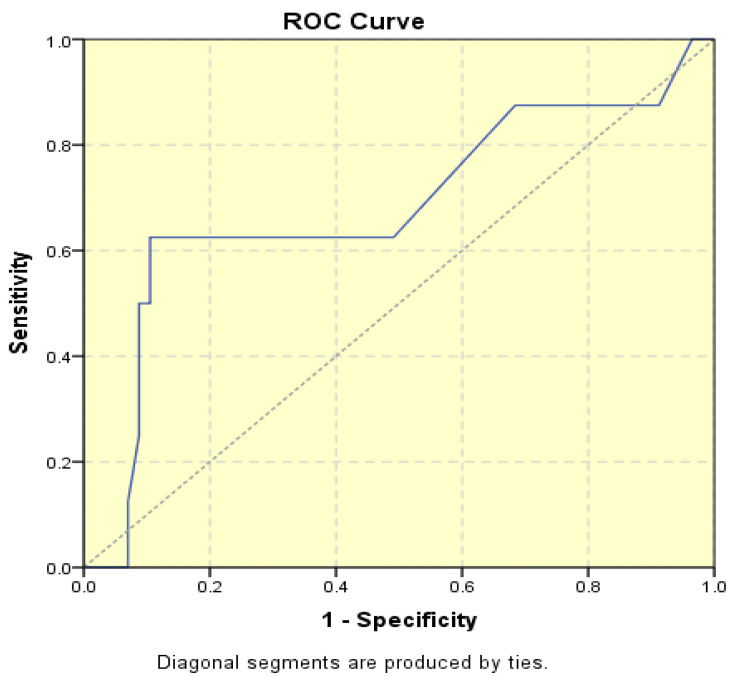
Receiver operating characteristic (ROC) curve illustrating the area under curve (AUC) for endometrial thickness in relation to malignancy.

**Figure 5 diagnostics-15-00891-f005:**
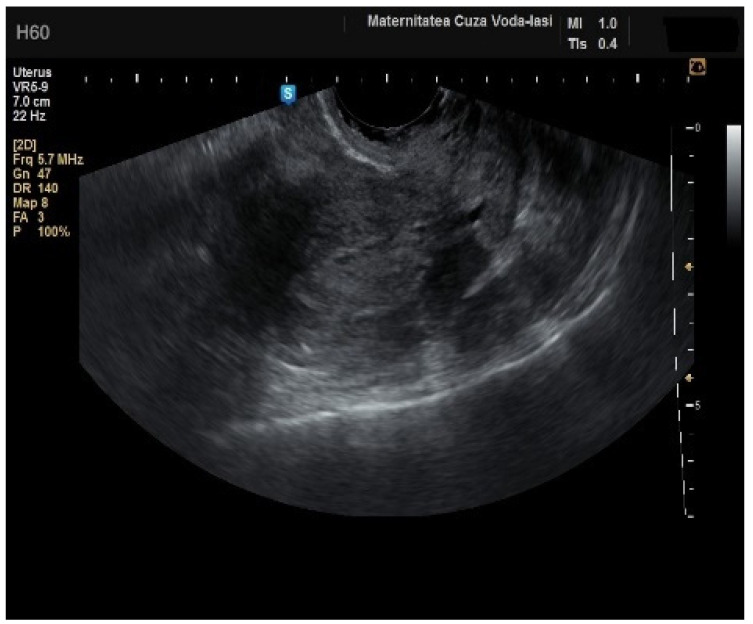
Heterogenous echogenicity.

**Figure 6 diagnostics-15-00891-f006:**
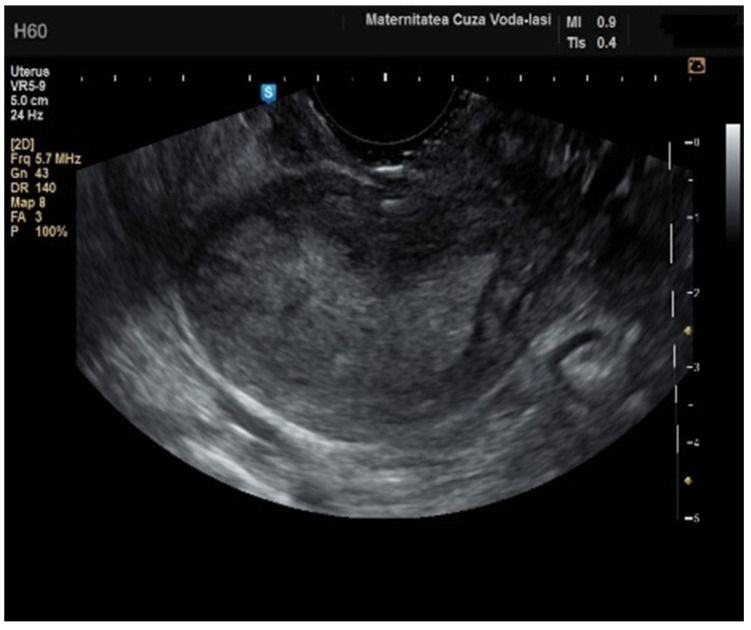
Irregular endometrial–myometrial junction.

**Figure 7 diagnostics-15-00891-f007:**
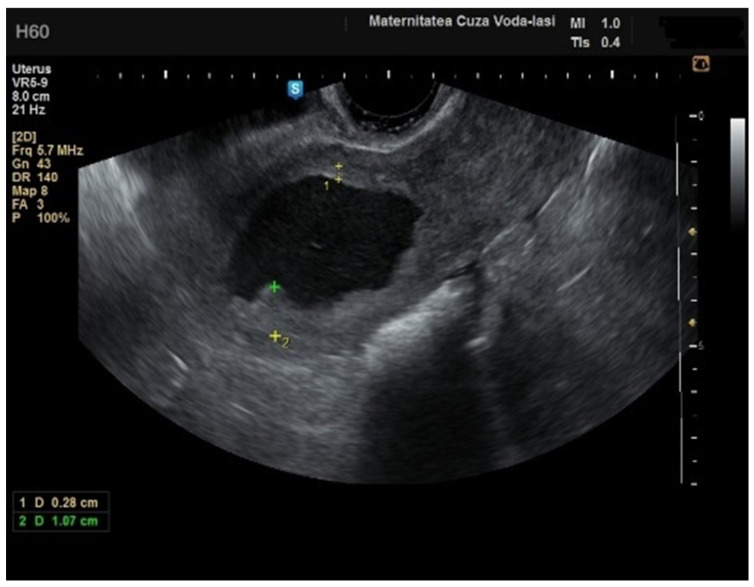
Irregular endometrial–myometrial junction with good visualization of the intrauterine tumor due to blood in the uterine cavity.

**Figure 8 diagnostics-15-00891-f008:**
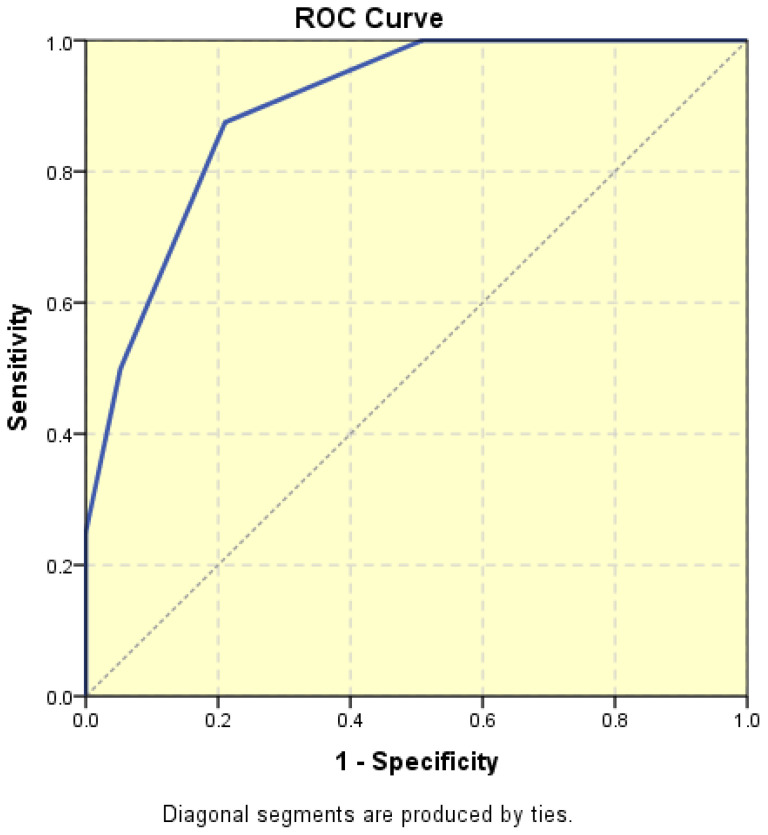
Receiver operating characteristic (ROC) curve illustrating the area under curve (AUC) for color Doppler vascular score 1 in endometrial cancer: the vascular score and its association with endometrial cancer.

**Figure 9 diagnostics-15-00891-f009:**
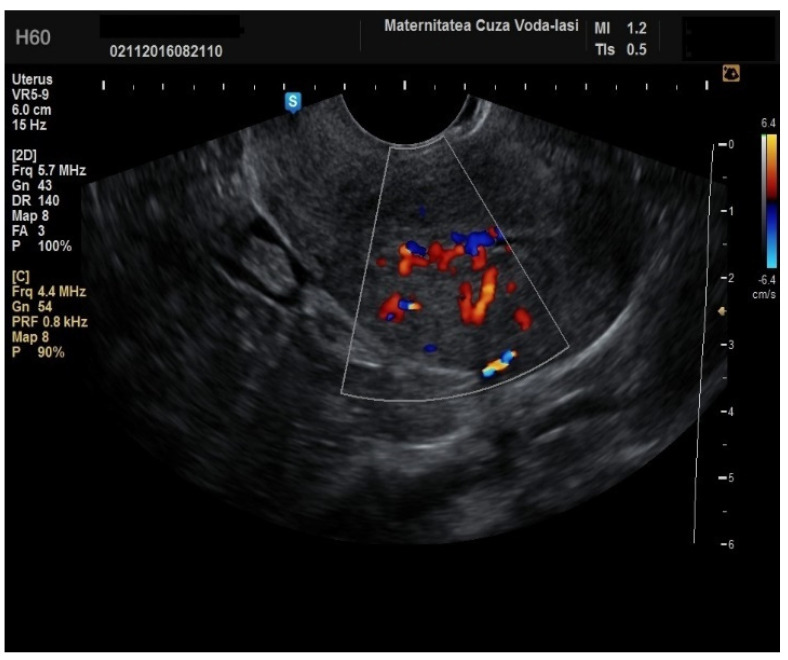
Color Doppler vascular score 4 in endometrial cancer.

**Figure 10 diagnostics-15-00891-f010:**
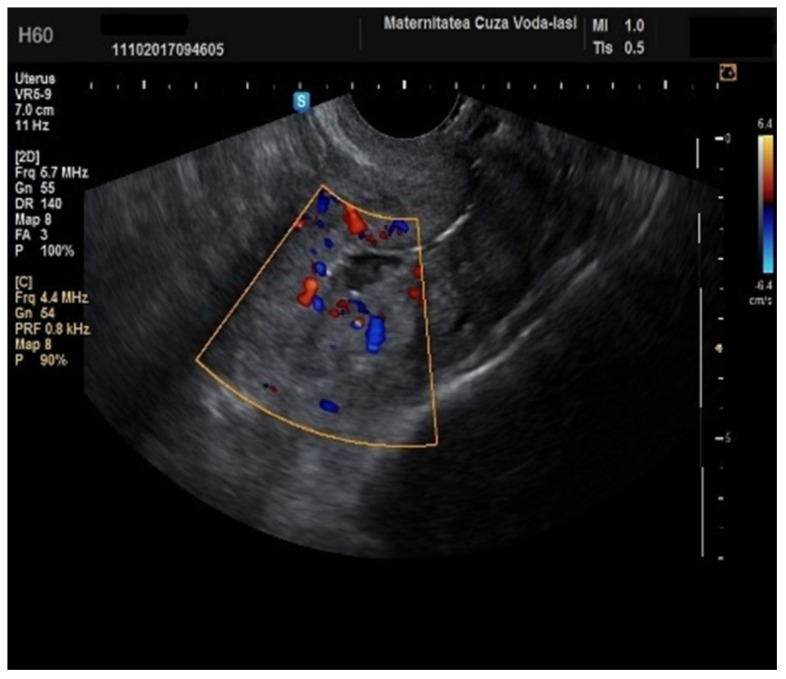
Color Doppler imaging for assessing endometrial invasion in endometrial cancer: scattered vascular patterns.

**Figure 11 diagnostics-15-00891-f011:**
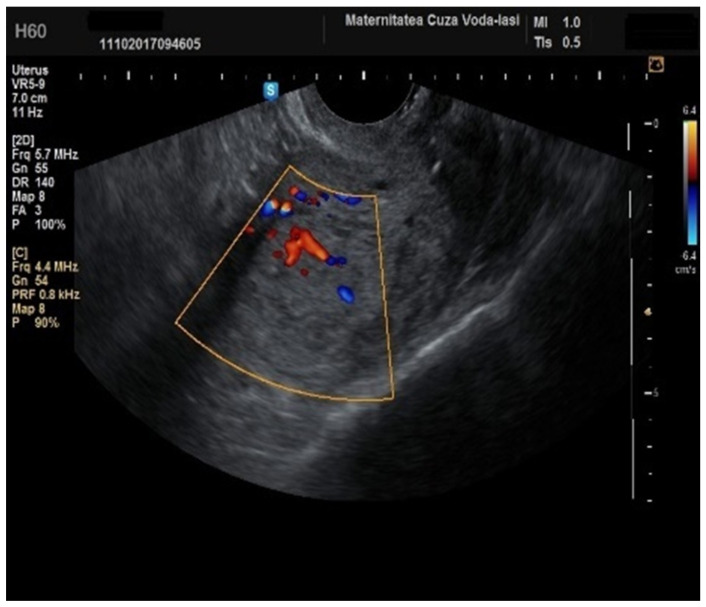
Color Doppler imaging for assessing endometrial invasion in endometrial cancer: presence of vessels with multiple branches.

**Figure 12 diagnostics-15-00891-f012:**
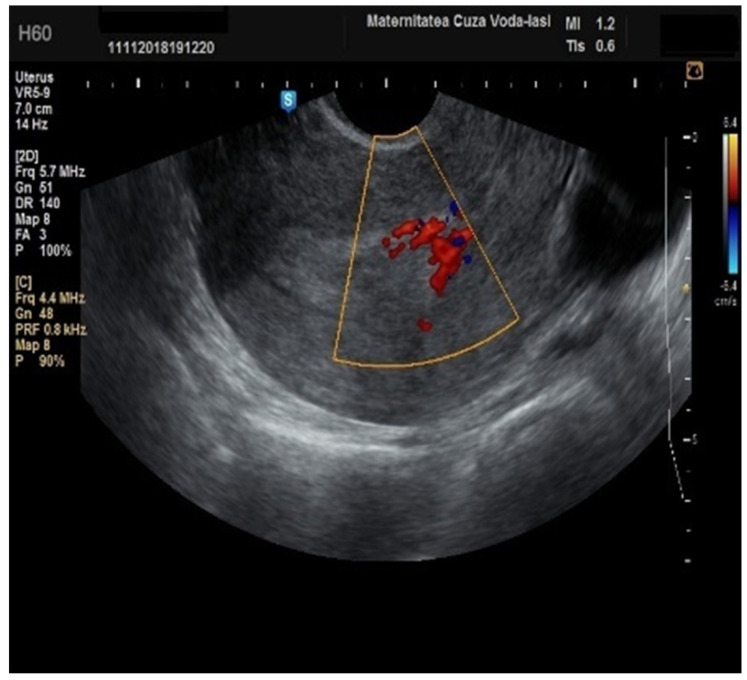
Color Doppler imaging for assessing endometrial invasion in endometrial cancer: presence of color splashes.

**Figure 13 diagnostics-15-00891-f013:**
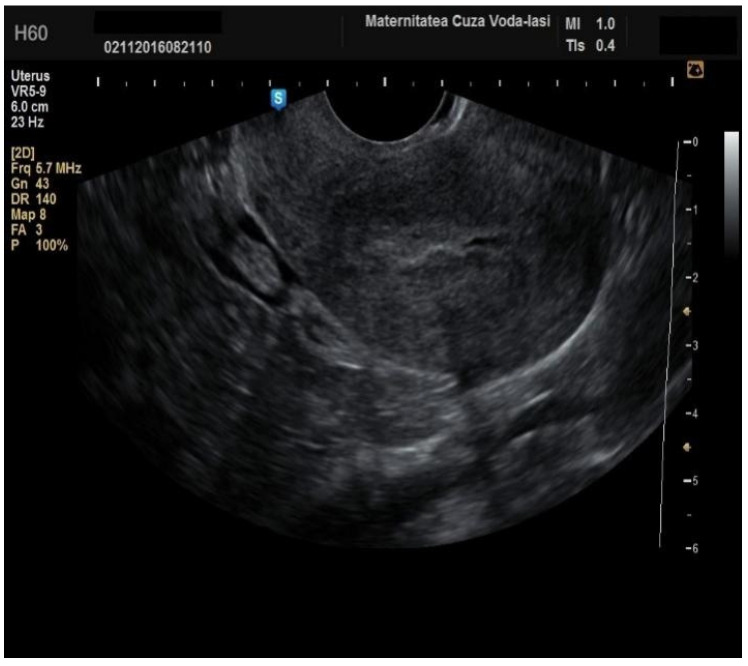
Ultrasound imaging in endometrial cancer: invasion of the posterior myometrium extending less than 50%—without vascularization.

**Figure 14 diagnostics-15-00891-f014:**
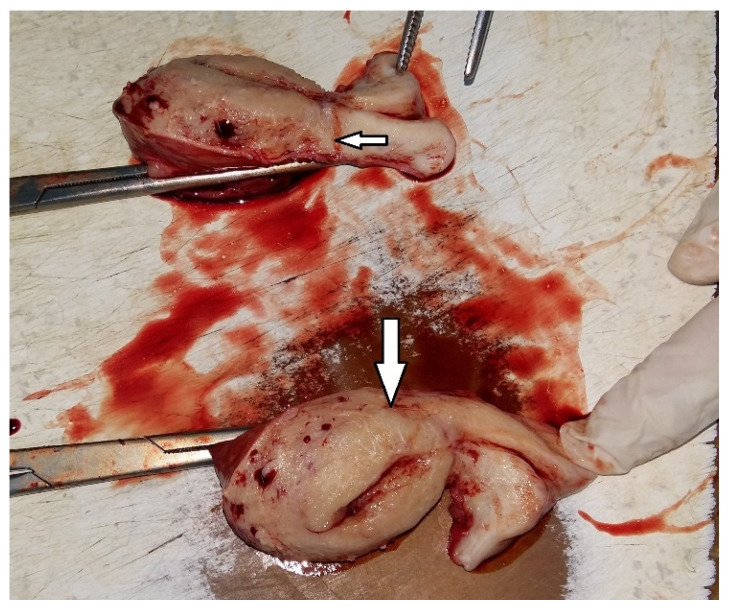
Macroscopic features of the case exhibiting invasion of the posterior myometrium, extending to less than 50% of its thickness (arrows).

**Figure 15 diagnostics-15-00891-f015:**
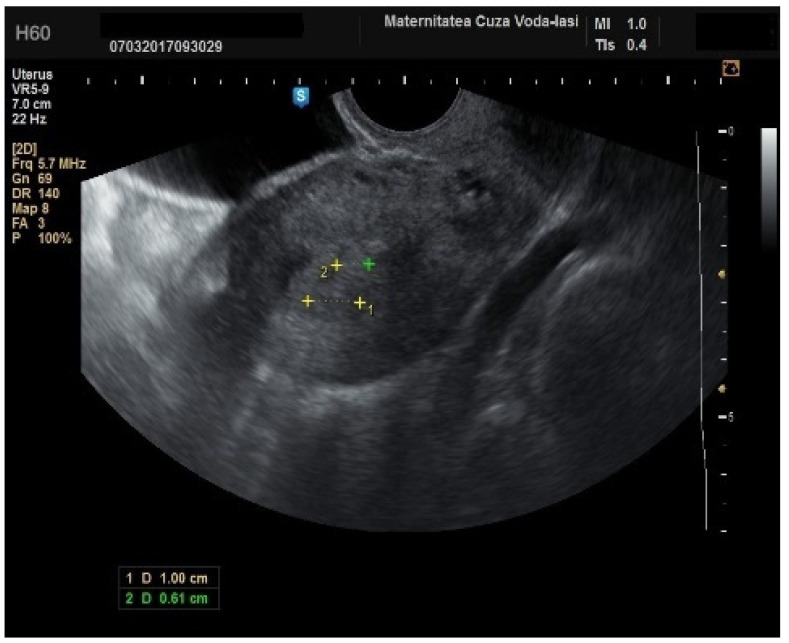
Endometrial polyp without vascularization.

**Figure 16 diagnostics-15-00891-f016:**
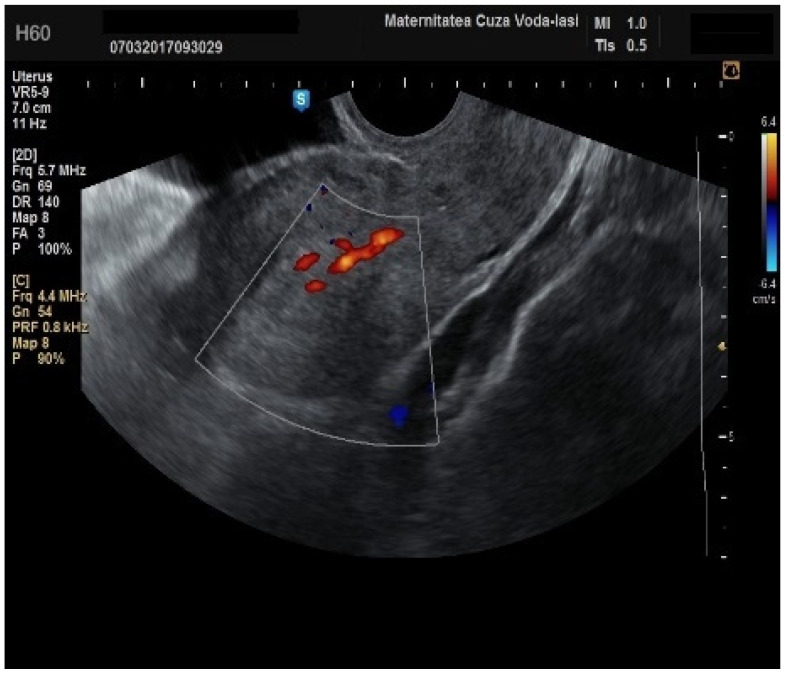
Color Doppler imaging for assessing endometrial polyps: single vessel with branching.

**Figure 17 diagnostics-15-00891-f017:**
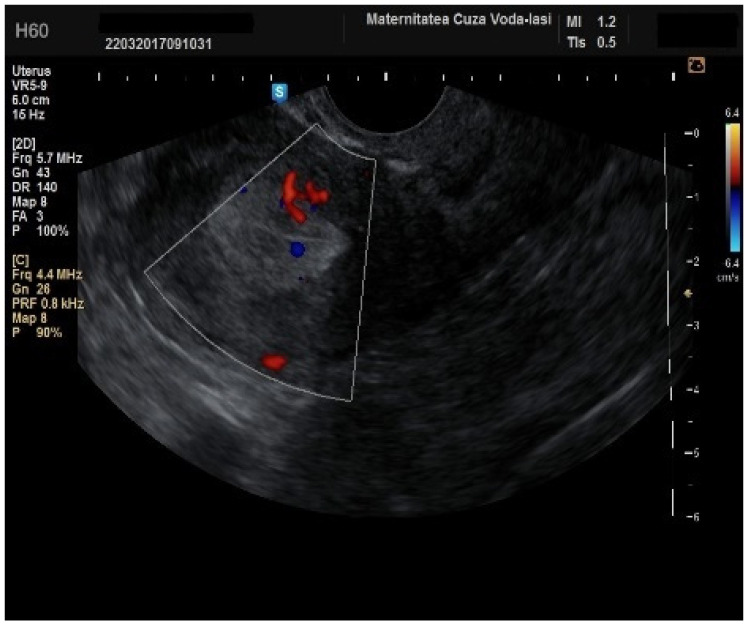
Color Doppler imaging for assessing endometrial polyps: single vessel without branching.

**Figure 18 diagnostics-15-00891-f018:**
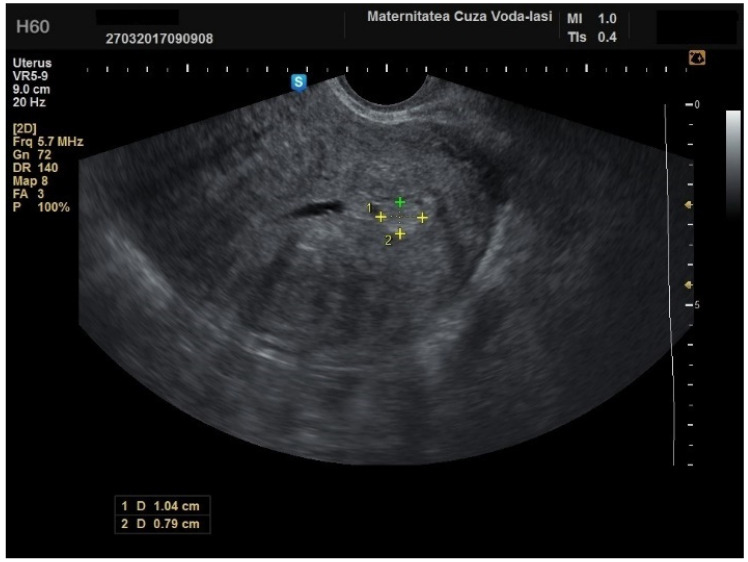
Endometrial polyp.

**Figure 19 diagnostics-15-00891-f019:**
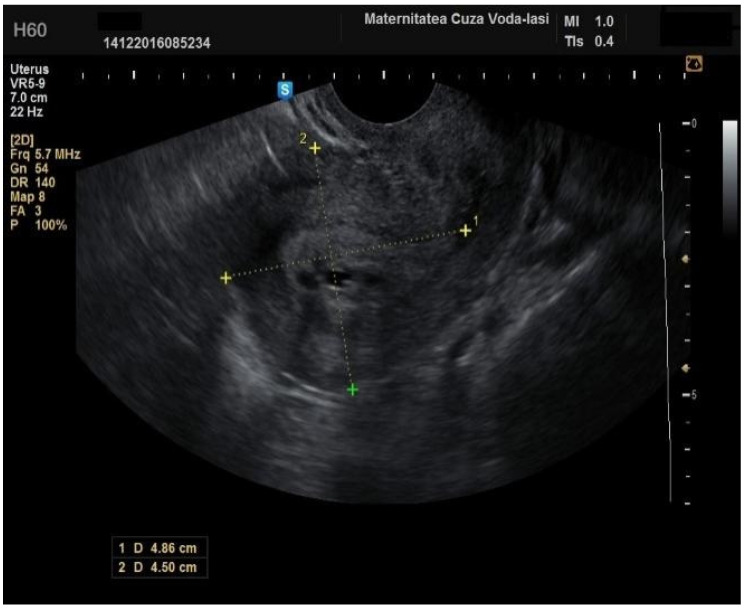
Cystic atrophic endometrium.

**Figure 20 diagnostics-15-00891-f020:**
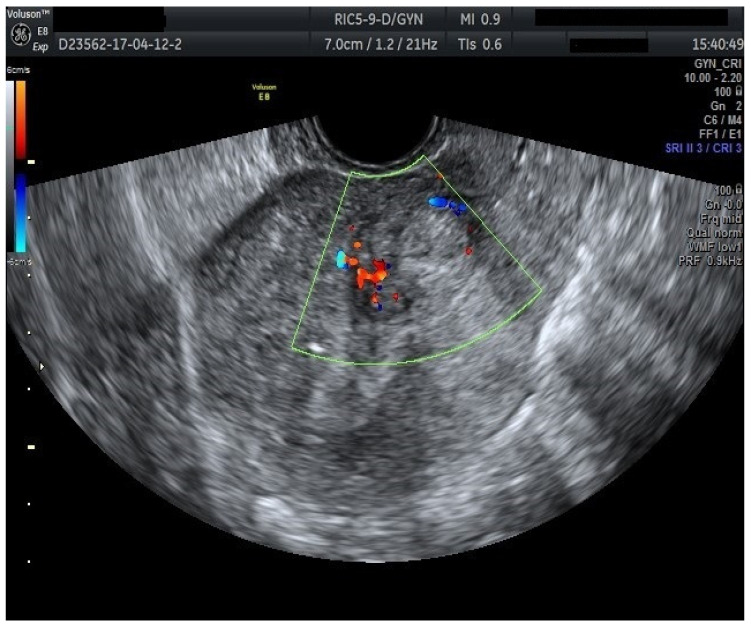
Myoma with circular flow pattern.

**Figure 21 diagnostics-15-00891-f021:**
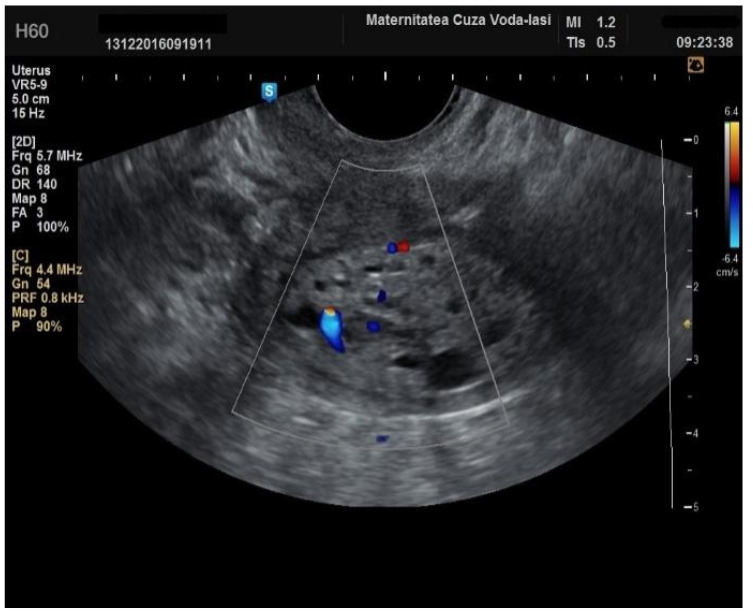
Cystic atrophic endometrium (Doppler examination).

**Table 1 diagnostics-15-00891-t001:** Characteristics of patients with endometrial cancer compared with those with non-cancer intracavitary pathology.

Characteristics	Endometrial Cancer		*p* Value
	Yes (*n* = 37)	No (*n* = 125)	
Menopause	30 (81.1%)	56 (44.8%)	0.001 ^(b)^
Mean age, years	61.97 ± 10.46	54.83 ± 10.44	0.001 ^(a)^
Obesity	15 (40.5%)	34 (27.2%)	0.127 ^(b)^
Symptomatic (vaginal bleeding)	37 (100%)	109 (87.2%)	0.003 ^(b)^
Diabetes	4 (10.8%)	6 (4.8%)	0.209 ^(b)^
Hypertension	7 (18.9%)	24 (19.2%)	0.970 ^(b)^
Endometrial thickness (mm)	24.49 ± 13.33	16.10 ± 9.37	0.001 ^(a)^
Endometrial uniformity	12 (32.4%)	38 (30.4%)	0.815 ^(b)^
Hyperechogenic aspect	35 (94.6%)	91 (72.8%)	0.006 ^(c)^
Irregular junction	37 (100%)	48 (34.3%)	0.001 ^(b)^
Color score	3 ± 1	1 ± 1	0.001 ^(a)^
Single vessel	2 (5.4%)	20 (16.0%)	0.072 ^(b)^
Multiple vessels with focal origin	0 (0%)	22 (17.6%)	0.001 ^(b)^
Branching vessels	2 (5.4%)	6 (4.8%)	0.883 ^(b)^
Circular flow	0 (0%)	5 (4.0%)	0.104 ^(b)^
Scattered vessels	20 (54.1%)	12 (9.6%)	0.001 ^(b)^

^(a)^ Student’s *t*-test; ^(b)^ Chi-square test; ^(c)^ Kruskal–Wallis tests.

## Data Availability

The authors affirm that the data supporting the findings of this study are available within the article. Raw data underpinning the results of this study may be obtained from the corresponding author, upon reasonable request.

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
