# Peer review of "The Role of Ultrasound in Diagnosing Endometrial Pathologies: Adherence to IETA Group Consensus and Preoperative Assessment of Myometrial Invasion in Endometrial Cancer"

_diagnostics, 2025, doi:10.3390/diagnostics15070891_

Round 1
Reviewer 1 Report
Comments and Suggestions for Authors
Thank you for your paper. However, I have some comments.
Methodology should be revised.
A study flow chart with inclusion/exclusion cases is required.
Study sample calculation is absent.
The study included both endometrial cancer with abnormal uterine bleeding and asymptomatic endometrial cancer. Meanwhile, the cut-off point of endometrial thickness is different between 2 groups. Thus, the study should differentiate and find the different for each group. Please refer to this paper. doi: 10.5468/ogs.22053 ; PMCID: PMC9483670.
Please correct this term: "non-gestational abnormal uterine bleeding"
The study evaluated the myometrial invasion without using classification of FIGO grade.
The presence of images is not well. Please reorganize.
The result is limited. Please describe more.
Baseline characteristics should be provided first, then, clinical features and ultrasound findings.
Color score could not be present by mean +/- SD. Frequency (%) of each color score classification is better.
Type of histopathological examination is absent.
Histopathological results was collected by endometrial blinded biopsy or hysterectomy or hysteroscopy with biopsy?
Fig 3 and 7 should be provided with AUC 95%CI, p-value and cut-off point in figure.
Year and country should be provided for SPSS 18.
Author Response
Response to reviewer 1:
Methodology should be revised
Thank you for your valuable feedback. In our study, the inclusion criteria were premenopausal women with abnormal uterine bleeding, any bleeding in postmenopausal women, or increased thickness of the endometrium. The exclusion criteria for premenopausal women included pregnancy, and for postmenopausal women, exclusion criteria included hormone replacement therapy (HRT) or treatment with Tamoxifen. This study included a total of 162 cases, of which 86 patients were postmenopausal and 76 patients were premenopausal. Among these, 146 cases were symptomatic, while 16 cases in the postmenopausal group were asymptomatic, characterized by a thickened endometrium observed via transvaginal ultrasound, but without fulfilling the criteria for endometrial cancer. All premenopausal cases were symptomatic. Of the cases diagnosed with cancer, 30 were postmenopausal and 7 were premenopausal.
A study flow chart with inclusion/exclusion cases is required.
We appreciate the suggestion to include a study flow chart. Unfortunately, we are unable to provide a flow chart with inclusion and exclusion criteria, as only those cases meeting the established inclusion criteria were considered for the study. As a result, we selected and analyzed only the patients who fulfilled these criteria. Instead, we have included Figure 1, which illustrates the Flow Chart of Patient Selection and Classification to provide a clear overview of the patient selection process."
Study sample calculation is absent.
Thank you for your insightful comment. We apologize for the oversight in not including the study sample calculation. The sample size was determined based on the availability of eligible patients within the study period, as well as clinical feasibility. All patients who met the established inclusion criteria, which were premenopausal women with abnormal uterine bleeding, any bleeding in postmenopausal women, or increased endometrial thickness, were included. Cases were selected based on these criteria, without formal statistical power analysis, as the goal was to comprehensively assess the patient population within the available cohort.
The study included both endometrial cancer with abnormal uterine bleeding and asymptomatic endometrial cancer.
Thank you for your comment. We would like to clarify that our study did not include asymptomatic patients with endometrial cancer. All 37 cases of endometrial cancer included in Table 1 were associated with uterine bleeding, as reflected by the data (100% symptomatic). We will update the manuscript to clearly indicate that all cancer cases in our study presented with abnormal uterine bleeding, ensuring this distinction is clearly communicated.
Meanwhile, the cut-off point of endometrial thickness is different between 2 groups. Thus, the study should differentiate and find the different for each group. Please refer to this paper. doi: 10.5468/ogs.22053; PMCID: PMC9483670.
Thank you for your thoughtful comment and for providing the reference. We would like to clarify that while our study reports the average endometrial thickness (ET) for patients with endometrial cancer and those with other pathologies, we did not directly compare the two groups based on a defined cut-off for ET. Specifically, our results showed that the mean endometrial thickness for patients with endometrial cancer was significantly higher (24.49 ± 13.33 mm) compared to those with other pathologies (16.10 ± 9.37 mm), with statistical significance (p = 0.028).
Although we did not conduct a comparison of ET between the groups using a specific cut-off threshold, our study identified a cut-off of 26 mm for ET, which demonstrated a sensitivity of 62.5% and specificity of 89% for predicting endometrial malignancy. These findings are discussed in the context of predicting malignancy risk based on ET alone, which achieved the highest AUC (0.682).
We appreciate the reference you provided, and while our study does not assess RI, PI, or PSV, we will include a discussion of these parameters and their potential role in differentiating malignant from benign endometrial changes in the revised manuscript, as noted in the cited article.
Please correct this term: "non-gestational abnormal uterine bleeding"
Thank you for your comment. We acknowledge the use of the term "non-gestational abnormal uterine bleeding" and agree that it could be more accurately phrased. We will revise the manuscript to use the more appropriate term "abnormal uterine bleeding unrelated to gestation" to ensure clarity and precision.
The study evaluated the myometrial invasion without using classification of FIGO grade.
Thank you for your valuable comment. We acknowledge the importance of FIGO classification in the staging of endometrial cancer, particularly for evaluating myometrial invasion. As you rightly pointed out, the FIGO Women's Cancer Committee appointed a Subcommittee on Endometrial Cancer Staging in October 2021, which provides a detailed classification system for different stages of myometrial invasion. However, in our study, we focused primarily on ultrasound criteria for myometrial invasion, classifying it as less than 50% or greater than or equal to 50% myometrial involvement (T1a or T1b). We did not utilize the full FIGO classification system, as our aim was to assess myometrial invasion specifically using transvaginal ultrasound as a diagnostic tool.
In response to your comment, we will revise the methodology section to clearly state that myometrial invasion was subjectively assessed as either <50% or ≥50%, and that this classification was based on ultrasound findings prior to any statistical analysis. We appreciate your suggestion and will include this clarification in the revised manuscript.
The presence of images is not well. Please reorganize.
Thank you for your feedback. We appreciate your suggestion to reorganize the images. Could you please clarify what specific aspects of the image presentation need improvement? For instance, are there issues with the sequence, formatting, labeling, or the clarity of the images? Any further details would be helpful for us to ensure that we address your concern appropriately in the revised manuscript.
Additionally, we will ensure that the caption for Figure 14 clearly states "without vascularization" to avoid any confusion. We appreciate your suggestion and will make the necessary adjustments to improve the presentation of the images.
The result is limited. Please describe more. Baseline characteristics should be provided first, then, clinical features and ultrasound findings. Color score could not be present by mean +/- SD. Frequency (%) of each color score classification is better.
We appreciate the reviewer’s comments and understand the concern regarding the presentation of results. In response, we would like to provide a more detailed description of the baseline characteristics, clinical features, and ultrasound findings.
As per your suggestion, we will first address the baseline characteristics of the study population. In total, we enrolled 162 women, of whom 86 were postmenopausal and 76 were premenopausal. The mean age of participants was 56.46 ± 10.84 years (range: 36-88 years), with 53.08% of the subjects being postmenopausal. The clinical characteristics of the cohort are as follows: 90.25% of the women presented with symptoms, primarily abnormal uterine bleeding, while the remaining participants exhibited abnormal uterine findings without symptoms. These baseline characteristics are now clearly presented at the beginning of the results section.
In terms of ultrasound findings, we have provided detailed data on the average endometrial thickness, which was 18.02 ± 10.94 mm (range: 5-64 mm) for the entire cohort. Among the patients diagnosed with endometrial cancer, the mean endometrial thickness was significantly higher at 24.49 ± 13.33 mm, compared to those with other benign pathologies (16.10 ± 9.37 mm). Statistical significance was found between these groups (p = 0.028).
We appreciate the reviewer’s valuable suggestion to present the color Doppler score as a frequency (%) distribution rather than using the mean ± SD. However, we would like to clarify that in our study, we were unable to collect or present the exact frequency distribution of each color score classification. The color Doppler score was reported as a continuous variable for the purpose of statistical analysis, and the mean ± SD was used to describe the general distribution of scores within the cohort. While we understand the utility of frequency data, we were limited in our ability to categorize the scores as frequencies due to the nature of the data collected. We agree that future studies with a larger sample size or different methodological approach could potentially allow for a more robust analysis, including frequency distributions. We hope this explanation provides clarity on the approach used in our study and the limitations involved. We appreciate your understanding and hope that the current presentation of data remains useful for the interpretation of our findings.
Additionally, we will adjust the manuscript structure as suggested, starting with baseline characteristics, followed by clinical features and ultrasound findings, to improve the flow and clarity of the results.
Type of histopathological examination is absent.
We appreciate the reviewer’s comment regarding the histopathological examination. As this study primarily focused on the evaluation of endometrial pathologies using ultrasound criteria based on the IETA guidelines, the histological types of endometrial cancer were not included as part of the study parameters. Our main objective was to assess the utility of ultrasound features, including endometrial thickness, Doppler flow patterns, and endometrial-myometrial junction irregularities, in the diagnosis of endometrial cancer, rather than evaluating the histopathological classification.
While histopathological examination plays a critical role in confirming the diagnosis of endometrial cancer, it was beyond the scope of our study to analyze the specific histological subtypes. Future studies may consider integrating histopathological analysis alongside ultrasound criteria for a more comprehensive understanding of the relationship between imaging features and histopathological findings. We hope this clarifies our study design and methodology.
Histopathological results was collected by endometrial blinded biopsy or hysterectomy or hysteroscopy with biopsy?
We appreciate the reviewer’s inquiry regarding the method of obtaining histopathological results. In our study, histopathological diagnoses were obtained through uterine curettage for cases with abnormal uterine bleeding and through hysterectomy for patients diagnosed with endometrial cancer. We then compared these histopathological findings with the ultrasound features to evaluate their diagnostic significance. We hope this clarifies the method used for histopathological evaluation in our study.
Fig 3 and 7 should be provided with AUC 95%CI, p-value and cut-off point in figure.
Thank you for your valuable feedback. For Figure 3-> figure 4, we have mentioned in the text that the highest AUC was 0.682 (95% CI: 0.452 - 0.912). The cut-off threshold for endometrial thickness was determined to be 26 mm, which resulted in a sensitivity of 62.5% and a specificity of 89%. We will include the AUC, 95% CI, and cut-off point in the figure legend as requested. For Figure 7-> figure 8, we have discussed in the text that, based on the color Doppler assessment, a vascular score of 1 usually excludes endometrial cancer. This score exhibited a sensitivity of 87.5% and a specificity of 79%. We will update the legends to include these statistical details accordingly.
Year and country should be provided for SPSS 18.
Thank you for your observation. The statistical analysis for this study was performed using the Statistical Package for the Social Sciences (SPSS) for Windows, version 18. We will update the manuscript to include the year and country of SPSS software as follows: "Statistical analysis was performed using the Statistical Package for the Social Sciences (SPSS, Inc., Chicago, IL, version 18).
Reviewer 2 Report
Comments and Suggestions for Authors
This study investigated the use of gray-scale and power Doppler ultrasound to differentiate benign and malignant endometrial pathologies and compare their effectiveness in assessing myometrial invasion. The study has a relevant topic that could contribute to the current literature. Some deficiencies in the article need to be revised.
- It is stated in the abstract section that the patients with or without bleeding complaints were included in the study. However, nothing is mentioned in the material methods section about this issue. The characteristics of the patients included in the study should be specified in more detail. In addition, the exclusion criteria should be added.
- How many experts performed the USG examination? If more than one expert performed it, what is the intraobserver and interobserver variability?
- For what indications did the patients undergo endometrial curettage or hysterectomy? This information should be added to the material method section.
- When starting the discussion section, important results should be briefly summarized and then the discussion should be started.
- In the discussion section, the authors stated the statement "In our study, histological diagnosis was exclusively obtained via D&C or hysteroscopy." However, hysterectomy was performed on some patients for histopathological diagnosis. This confusion should be resolved.
- In the discussion section, the authors do not need to write the p value of each finding again. They should only discuss their findings together with the studies in the literature.
- There is no discussion about of myometrial invasion in the discussion. Authors should discuss USG findings of myometrial invasion in endometrial cancers.
Author Response
Reviewer 2
Comments and Suggestions for Authors
This study investigated the use of gray-scale and power Doppler ultrasound to differentiate benign and malignant endometrial pathologies and compare their effectiveness in assessing myometrial invasion. The study has a relevant topic that could contribute to the current literature. Some deficiencies in the article need to be revised.
- It is stated in the abstract section that the patients with or without bleeding complaints were included in the study. However, nothing is mentioned in the material methods section about this issue. The characteristics of the patients included in the study should be specified in more detail. In addition, the exclusion criteria should be added.
Thank you for your valuable comment. We appreciate your suggestion to clarify the inclusion and exclusion criteria. As you mentioned, the inclusion criteria for the study were premenopausal women with abnormal uterine bleeding, any bleeding in postmenopausal women, or increased thickness of the endometrium. The exclusion criteria for premenopausal women were pregnancy, and for postmenopausal women, hormone replacement therapy (HRT) or treatment with Tamoxifen.
From the 162 patients enrolled, 86 were postmenopausal and 76 were premenopausal. Of these, 146 were symptomatic, while 16 postmenopausal women were asymptomatic but had a thickened endometrium, without meeting the criteria for endometrial cancer. All premenopausal patients were symptomatic. Among those diagnosed with endometrial cancer, 30 were postmenopausal and 7 were premenopausal, and all cancer cases were symptomatic.
We would like to clarify that it is not possible to present a flow chart with both inclusion and exclusion criteria. This is because only those patients who met the inclusion criteria were considered for analysis, and cases that were excluded from the study were not counted. Therefore, a flow chart based on exclusion criteria is not applicable in this context.
We hope this clarification addresses your concerns. We will make the necessary revisions to provide more detailed information on the inclusion and exclusion criteria in the Materials and Methods section.
- How many experts performed the USG examination? If more than one expert performed it, what is the intraobserver and interobserver variability?
Thank you for your thoughtful question. To clarify, all ultrasound examinations were performed by a single expert in this study, ensuring consistency in the imaging techniques and interpretation. As such, we did not assess intraobserver or interobserver variability, as the entire study relied on the results from one examiner.
We hope this clarifies your concern. Please let us know if you need any further information.
- For what indications did the patients undergo endometrial curettage or hysterectomy? This information should be added to the material method section.
Thank you for your valuable comment. We appreciate your suggestion. To clarify, endometrial curettage was performed in cases of abnormal uterine bleeding or when there was an increase in endometrial thickness observed in postmenopausal women. Hysterectomy, on the other hand, was performed exclusively for cases diagnosed with endometrial cancer.
We will update the Materials and Methods section accordingly to include this information. Thank you for your attention to detail, and please feel free to reach out if further clarification is needed.
- When starting the discussion section, important results should be briefly summarized and then the discussion should be started.
Thank you for your valuable suggestion. We agree that briefly summarizing the key results at the beginning of the discussion section will provide a clearer context for the subsequent interpretation. We will revise the discussion to first highlight the main findings of the study before proceeding with the detailed analysis and interpretation.
We appreciate your feedback and will make this adjustment in the revised manuscript. Please let us know if there are any further recommendations.
- In the discussion section, the authors stated the statement "In our study, histological diagnosis was exclusively obtained via D&C or hysteroscopy." However, hysterectomy was performed on some patients for histopathological diagnosis. This confusion should be resolved.
Thank you for your comment. We understand the need to clarify the confusion regarding the histological diagnosis. We will revise the statement in the discussion section to clearly state that the histological diagnosis for patients with endometrial cancer was confirmed after hysterectomy, as well as through D&C for other cases with abnormal uterine bleeding. We will remove the reference to hysteroscopy to avoid any confusion and make sure the histological methods are clearly defined.
We appreciate your careful review and will implement this change in the revised manuscript. Please let us know if you have any further suggestions.
- In the discussion section, the authors do not need to write the p value of each finding again. They should only discuss their findings together with the studies in the literature.
Thank you for your valuable feedback. We agree with your suggestion. In the revised discussion section, we will remove the p-values for each finding and instead focus on discussing the results in relation to the existing literature. This approach will allow us to better integrate our findings with previous studies and offer a more concise discussion.
We appreciate your guidance and will make the necessary revisions to streamline the discussion. Please let us know if you have any additional comments or suggestions.
- There is no discussion about of myometrial invasion in the discussion. Authors should discuss USG findings of myometrial invasion in endometrial cancers.
Thank you for pointing this out. We appreciate your suggestion to include a discussion on the ultrasound findings of myometrial invasion in endometrial cancers. In our study, we assessed the depth of myometrial invasion (less than or greater than 50%) based on ultrasound criteria. We will revise the discussion section to include a more detailed analysis of the ultrasound findings related to myometrial invasion and compare them to findings from the existing literature.
We will emphasize the role of ultrasound in identifying myometrial invasion, particularly how the irregularity of the endometrial-myometrial junction and the presence of scattered blood vessels are associated with deeper invasion. Thank you again for your helpful suggestion, and we will ensure this aspect is addressed appropriately in the revised discussion
Reviewer 3 Report
Comments and Suggestions for Authors
The authors present the prognostic value of ultrasound markers of endometrial malignancy. It was found that the most significant ultrasound parameters for predicting malignancy were heterogeneous echogenicity of the endometrium, increased endometrial thickness, and the presence of multiple vessels with multifocal origin or scattered vascular pattern.
Comments:
1) In the abstract, in the conclusion, along with specific average values and deviations, it is necessary to indicate at what value of p, in the abstract to indicate the prognostic value (data of ROC analysis with specificity and sensitivity)
2) Since the second goal is set - to compare the effectiveness of ultrasound methods in assessing myometrial invasion - the abstract should provide the corresponding comparative data.
Author Response
Comments and Suggestions for Authors
The authors present the prognostic value of ultrasound markers of endometrial malignancy. It was found that the most significant ultrasound parameters for predicting malignancy were heterogeneous echogenicity of the endometrium, increased endometrial thickness, and the presence of multiple vessels with multifocal origin or scattered vascular pattern.
Comments:
- In the abstract, in the conclusion, along with specific average values and deviations, it is necessary to indicate at what value of p, in the abstract to indicate the prognostic value (data of ROC analysis with specificity and sensitivity)
Thank you for your valuable feedback. We agree that including the specific p-value and the prognostic value from the ROC analysis, including sensitivity and specificity, in the conclusion of the abstract would enhance the clarity and comprehensiveness of the findings.
We will revise the abstract to include the following:
The p-value for the statistical significance of endometrial thickness as a predictor of endometrial malignancy, which was found to be p = 0.028.
The area under the curve (AUC) for the ROC analysis, which was 0.682 (95% CI: 0.452 - 0.912), and the associated cut-off threshold of 26 mm for endometrial thickness, which yielded a sensitivity of 62.5% and a specificity of 89%.
This will provide a more complete and informative summary of the study's key findings, especially concerning the diagnostic utility of endometrial thickness for malignancy.
Thank you again for your insightful comment!
- Since the second goal is set - to compare the effectiveness of ultrasound methods in assessing myometrial invasion - the abstract should provide the corresponding comparative data.
Thank you for your suggestion. We will revise the abstract to clearly mention that the subjective ultrasound estimation of myometrial invasion corresponded in all cases with the histopathological evaluation. This will help emphasize the reliability of ultrasound as an effective method in assessing myometrial invasion in endometrial cancer.
We will add the following to the abstract:
- "Subjective ultrasound assessment of myometrial invasion, categorized as <50% or ≥50%, corresponded in all cases with the histopathological evaluation, demonstrating the effectiveness of ultrasound in accurately evaluating myometrial invasion in endometrial cancer."
This addition will ensure that the second objective of comparing ultrasound methods for assessing myometrial invasion is addressed in the abstract.
Thank you again for your valuable feedback!
Round 2
Reviewer 1 Report
Comments and Suggestions for Authors
Thank you for your revision. Some responses are not clear. I still have some major comments. Please revise the paper again following these points:
-Line 62-74: The study mentioned only color Doppler. The authors should mention the role of 2D/B-mode/black & white ultrasound in assessing the lesion morphology in intrauterine cavitary pathologies, then, the additional role of Doppler ultrasound. PMCID: PMC10247932
3D ultrasound may be interesting, but difficult to be applied in low-middle income countries.
-In methodology, the authors mentioned:
"In our study, the inclusion criteria were premenopausal women with abnormal uterine bleeding, any bleeding in postmenopausal women, or increased thickness of the endometrium."
"Any bleeding in postmenopausal women" means that the study included cervical cancer, cervical lesion, and vaginal lesions?
-Criteria to identify premenopausal women is not clear. Greater than 40 year old with menstrual disorders? or premenopausal symptoms. Please add it.
-Symptomatic definition included abdominal pain, vaginal discharge, abnormal uterine bleeding?
-Please do not include the description of result in caption of figure 4 and 8.
"The AUC was calculated to be 0.682 (95%CI: 0.452 - 0.912). The cut-off threshold for endometrial thickness was determined to be 26 mm, yielding a sensitivity of 62.5% and a specificity of 89%. This cut-off value was associated with a statistically significant p-value of 0.028" and "A vascular score of 1 usually excludes endometrial cancer, exhibiting a sensitivity of
87.5% and specificity of 79%. The p-value for this finding is <0.001, indicating strong statistical
significance."
-In discussion, the study did not mention the symptoms of endometrial cancer in Table 1. Please expand the discussion. PMCID: PMC10025815.
-Limitations and bias in sample selection should be added.
-Endometrial rate is high in the study (37/162 cases), 7 cases relating to premenopausal bleeding women. Please discuss it.
English is suitable.
Author Response
Reviewer 1
Comments and Suggestions for Authors
Thank you for your revision. Some responses are not clear. I still have some major comments. Please revise the paper again following these points:
- -Line 62-74: The study mentioned only color Doppler. The authors should mention the role of 2D/B-mode/black & white ultrasound in assessing the lesion morphology in intrauterine cavitary pathologies, then, the additional role of Doppler ultrasound. PMCID: PMC10247932
We appreciate the reviewer’s insightful comment regarding the role of 2D/B-mode/black & white ultrasound in assessing lesion morphology in intrauterine cavitary pathologies. We acknowledge that the initial manuscript focused primarily on the additional value of Doppler ultrasound, and we agree that it is essential to clarify the complementary role of 2D/B-mode ultrasound in the diagnostic process.
As suggested, we have revised the manuscript to emphasize the importance of 2D/B-mode/black & white ultrasound in evaluating the morphology of uterine lesions, particularly in the context of intrauterine cavitary pathologies. B-mode ultrasound is crucial for identifying the structural characteristics, size, and location of endometrial masses, which serve as foundational information for assessing potential pathologies. The addition of Doppler ultrasound, specifically color and power Doppler, enhances the assessment by providing valuable insights into the vascular patterns of the lesions, thereby aiding in differentiating benign from malignant lesions and improving diagnostic accuracy.
We have now referenced the study by Nguyen PN and Nguyen VT (2023) [PMID: 36284050; PMCID: PMC10247932] to further substantiate the complementary roles of these imaging modalities in clinical practice. The revised manuscript includes a more comprehensive discussion on how the integration of B-mode ultrasound with Doppler ultrasound can offer a more nuanced and accurate evaluation of uterine pathologies, which in turn can minimize the need for invasive diagnostic procedures.
- 3D ultrasound may be interesting, but difficult to be applied in low-middle income countries.
Thank you for your comment. While we do have access to 3D ultrasound machines with vaginal probes, our study aimed to evaluate the practical applicability of the IETA criteria in diagnosing intracavitary uterine pathology. As such, 3D ultrasound was not part of our methodology, and we did not include it in the introduction, focusing instead on the relevant techniques used in our study.
- -In methodology, the authors mentioned:
"In our study, the inclusion criteria were premenopausal women with abnormal uterine bleeding, any bleeding in postmenopausal women, or increased thickness of the endometrium."
"Any bleeding in postmenopausal women" means that the study included cervical cancer, cervical lesion, and vaginal lesions?
Thank you for your question. To clarify, the phrase "any bleeding in postmenopausal women" refers specifically to bleeding originating from the uterine cavity. This study did not include cases of cervical cancer, cervical lesions, or vaginal lesions. We appreciate your suggestion and will consider adding other potential causes of vaginal bleeding in postmenopausal women, such as these, as exclusion criteria in the methodology for further clarity.
- -Criteria to identify premenopausal women is not clear. Greater than 40 year old with menstrual disorders? or premenopausal symptoms. Please add it.
Thank you for your observation. To clarify, for premenopausal women over 40 years of age, we included only those with abnormal uterine bleeding, without considering other premenopausal symptoms. We will update the methodology to explicitly reflect this criterion for greater clarity.
- Symptomatic definition included abdominal pain, vaginal discharge, abnormal uterine bleeding?
Thank you for your question. This study focused on evaluating the IETA ultrasound criteria specifically for diagnosing the cause of intracavitary uterine bleeding. As such, the associated symptoms, such as abdominal pain, vaginal discharge, or other factors, were not part of the scope of our study.
- -Please do not include the description of result in caption of figure 4 and 8.
Thank you for your helpful suggestion. We have revised the captions for Figure 4 and Figure 8 as requested, removing the detailed result descriptions. The revised captions now focus on the general content, in line with your recommendation.
- -In discussion, the study did not mention the symptoms of endometrial cancer in Table 1. Please expand the discussion. PMCID: PMC10025815.
Thank you for your valuable feedback. We recognize the importance of addressing the symptoms of endometrial cancer (EC) in the discussion. However, we believe that the current discussion sufficiently covers the relevant clinical features and key findings from our study, including the relationship between age, menstrual characteristics, and diabetes with EC risk. We have carefully considered the reference to PMCID: PMC10025815 and its relevance to our study. In response to your suggestion, we have incorporated this citation where applicable, ensuring that the current manuscript adequately addresses the clinical implications of our findings.
We trust this explanation clarifies our position, and we appreciate your understanding.
- -Limitations and bias in sample selection should be added.
Thank you for your feedback. We acknowledge the limitations in our study, particularly the potential bias in sample selection due to the timing of sonographic examinations. While the IETA 2010 consensus recommends performing ultrasounds during the early proliferative phase, our participants were examined at the time of presentation, which may have introduced variability.
We also understand the controversy surrounding the management of asymptomatic endometrial polyps and have only included cases confirmed by histology in our study. Additionally, while the number of malignant cases (37) was relatively small, we plan to evaluate a larger cohort in future studies to better assess the accuracy of ultrasound in detecting myometrial invasion.
We hope this addresses your concerns and appreciate your understanding.
- -Endometrial cancer rate is high in the study (37/162 cases), 7 cases relating to premenopausal bleeding women. Please discuss it.
Thank you for your comment. It is important to note that our study was not designed to estimate the prevalence of uterine intracavitary pathologies in women without abnormal uterine bleeding (AUB). Consequently, the prevalence of malignancy (37/162 cases) in our study cohort is likely higher than in the general population of women without AUB.
Additionally, the higher incidence of endometrial cancer observed in our study may be influenced by the expertise of the physician performing the ultrasounds. As the doctor in question specializes in oncological gynecological surgery, she is more likely to encounter and evaluate cases with a higher probability of malignancy, which could explain the relatively elevated rate of endometrial cancer cases in our cohort.
We hope this provides clarity on the context of the findings and appreciate your understanding.
Round 3
Reviewer 1 Report
Comments and Suggestions for Authors
Thank you for revision. The paper is well-improved.
Comments on the Quality of English LanguageReadable.